# Preparation and In Vitro Evaluation of Chitosan-g-Oligolactide Based Films and Macroporous Hydrogels for Tissue Engineering

**DOI:** 10.3390/polym15040907

**Published:** 2023-02-11

**Authors:** Tatiana Tolstova, Maria Drozdova, Tatiana Popyrina, Diana Matveeva, Tatiana Demina, Tatiana Akopova, Elena Andreeva, Elena Markvicheva

**Affiliations:** 1Shemyakin-Ovchinnikov Institute of Bioorganic Chemistry, Russian Academy of Sciences, 16/10 Miklukho-Maklaya Str., 117997 Moscow, Russia; 2Institute of Biomedical Chemistry, 10 bld. 8, Pogodinskaya Str., 119121 Moscow, Russia; 3Enikolopov Institute of Synthetic Polymeric Materials, Russian Academy of Sciences, 70 Profsouznaya Str., 117393 Moscow, Russia; 4Institute of Biomedical Problems, Russian Academy of Sciences, 76a Khoroshevskoe Shosse, 123007 Moscow, Russia

**Keywords:** chitosan, chitosan-g-oligo (L,L-/L,D-lactides), hydrogels, films, L929 mouse fibroblasts, mesenchymal stromal cells, tissue engineering

## Abstract

In the current study, novel matrices based on chitosan-g-oligo (L,L-/L,D-lactide) copolymers were fabricated. In particular, 2D films were prepared by solvent casting, while 3D macroporous hydrogels were obtained by lyophilization of copolymer solutions. Copolymers of chitosan (Chit) with semi-crystalline oligo (L,L-lactide) (Chit-LL) or amorphous oligo (L,D-lactide) (Chit-LD) were obtained by solid-state mechanochemical synthesis. The structure of the hydrogels was found to be a system of interconnected macropores with an average size of 150 μm. In vitro degradation of these copolymer-based matrices was shown to increase in the case of the Chit-LL-based hydrogel by 34% and decrease for the Chit-LD-based hydrogel by 23% compared to the parameter of the Chit sample. Localization and distribution of mouse fibroblast L929 cells and adipose tissue-derived mesenchymal stromal cells (MSCs) within the hydrogels was studied by confocal laser scanning microscopy (CLSM). Moreover, cellular response, namely cell adhesion, spreading, growth, proliferation, as well as cell differentiation in vitro were also evaluated in the hydrogels for 10–14 days. Both the Chit-LL and Chit-LD matrices were shown to support cell growth and proliferation, while they had improved swelling compared to the Chit matrix. Osteogenic MSCs differentiation on the copolymer-based films was studied by quantitative reverse-transcriptase polymerase chain reaction (qRT-PCR). Maximal expression levels of osteogenesis markers (alkaline phosphatase (*ALPL*), bone transcription factor (*Runx*2), and osteopontin (*SPP*1) were revealed for the Chit-LD films. Thus, osteodifferentiation was demonstrated to depend on the film composition. Both Chit-LL and Chit-LD copolymer-based matrices are promising for tissue engineering.

## 1. Introduction

As is well known, tissues damaged with small defects as a result of illness or injury are capable for self-regeneration. However, critical size damage can be repaired only by transplantation of one’s own or donor tissues and/or organs. Tissue engineering (TE) is a promising alternative approach to transplantation, as it overcomes the acute shortage of donors and reduces possible immune rejection [1]. One of the keystones in TE is the development of polymer matrices (scaffolds) that can effectively support the repair or regeneration of damaged tissues [2]. Scaffolds should provide cell attachment and growth, which means that they have to mimic properties of the natural extracellular matrix (ECM) of various tissues to be repaired. For example, scaffold folds for bone repair should have a rather high mechanical strength [3,4,5]. Matrices for skin regeneration could provide an antibacterial effect because of possible bacterial infection [6,7]. Scaffolds can be functionalized with various inorganic components, for example, hydroxyapatite [8] or graphene oxide [9] or different drugs [10], including antitumor therapeutics [11]. In addition, the matrices should provide a rather porous structure for the diffusion of oxygen and nutrients [12], while the biodegradation rate should be adjusted to the regeneration rate of the replaced tissue [13].

To fabricate scaffolds for tissue engineering, natural polymers are widely employed, since they are biocompatible, biodegradable, and non-cytotoxic [14]. Being a deacylated form of chitin, chitosan belongs to the natural polysaccharides widely distributed in nature. Positively charged chitosan-based matrices have been shown to support cell adhesion, growth and proliferation, as well as to exhibit antibacterial properties [15,16,17]. However, these matrices have rather poor mechanical properties [18,19] and rather low degradation rate, which depend on molecular weight (MW) and deacetylation degree (DD) [20]. The biodegradation of chitosan even with rather low MW and rather high DD is often insufficient for the regeneration of some tissues, for example, peripheral nerves, cartilage, and blood vessels [21,22].

To overcome these limitations, various Chit derivatives as well as copolymers of Chit with other monomers/oligomers could be used for matrix fabrication. Due to the presence of functional amino groups, the Chit molecule could bind to various groups and in this way, one could enhance mechanical strength, biodegradation, and bioactivity [23,24,25,26]. For instance, to increase hydrolytic and enzymatic decomposition of the matrix, chitosan-Ɛ-caprolactone dialdehyde copolymers have been proposed [27]. Luckachan and Pillai showed that grafting hydrophobic oligo-L-lactide onto chitosan allowed an enhancement of the biodegradation rate and swelling of copolymers [28]. Yan J. et al. reported an improvement of mechanical properties, namely a tensile strength of multicomponent cross-linked scaffolds from chitosan, hyaluronic acid, and collagen [29]. It has also been shown that an the introduction of polyhedral oligomeric silsesquioxanes into chitosan-based matrices improved the rheological and biological properties of the hybrid composites [5]. 

Being FDA approved, polylactic acid (PLA) is one of few synthetic polymers that is used clinically. Polylactic acid meets most of the requirements for biomaterials due to its biocompatibility and degradation as well as easy formation of PLA-based matrices with a desired morphology [30]. Nevertheless, PLA’s hydrophobic nature and an absence of functional groups leads to a lack of cell affinity and, as a result, limits the use of PLA scaffolds for tissue engineering. Non-modified PLA-based materials do not provide desirable cell adhesion to the matrix surface [31]. However, PLA seems to be promising for grafting onto chitosan backbone.

A combination of chitosan and polylactic acid may be of great interest for copolymer-based scaffolds production. On one hand, it allows for the enhancement of the bioadhesive properties of these matrices due to chitosan. On the other hand, it is possible to adjust a hydrophilic-hydrophobic balance as well as the mechanical properties of the copolymer matrices due to PLA. Moreover, chitosan-g-poly(D, L-lactic acid) copolymers provide controllable biodegradation and drug release [32]. In addition, a basic nature of chitosan can be used to neutralize the acidic degradation products of polylactide [33]. It should be also mentioned that PLA biodegradation rate depends on crystallinity, molecular weight, and stereoisomers content [34], and therefore it could be adjusted by copolymerization with Chit [26].

Recently, we have obtained chitosan-g-oligolactides using an original mechanochemical approach [35]. The grafting of oligolactide fragments onto chitosan expands a number of methods for fabrication of three-dimensional (3D) scaffolds from the copolymers, for instance it allows the use of both the traditional freeze-drying method [36] and the laser stereolithography technology [35]. It should be also noted that the copolymer-based matrices obtained by photocuring were shown to support growth and differentiation of neural stem cells [37,38].

In the current study, biodegradable matrices based on chitosan-g-oligo(L,L-/L,D-lactide) copolymers were produced. In particular, 2D films were prepared by pouring, while 3D macroporous hydrogels were obtained by lyophilization of copolymer solutions. For the first time, the swelling and biodegradation behavior of these copolymer-based matrices was shown to depend on the matrix composition, namely an oligolactide stereoisomer type. In addition, cellular response, namely cell adhesion, spreading, growth, and proliferation were evaluated for L929 mouse fibroblasts and human MSCs at long-term cultivation in these matrices. Moreover, for the first time, we studied differentiation of MSCs on the chitosan-g-oligo(L,L-/L,D-lactide) copolymer films.

The aim of the study was to obtain biodegradable matrices, in particular the films (2D) and the macroporous hydrogels (3D) from graft copolymers of chitosan with L,L-/L,D-oligolactides, to evaluate their structure and some physicochemical properties, and to test them in vitro.

## 2. Materials and Methods

### 2.1. Chemicals 

Dexamethasone, ascorbate-2-phosphate, beta-glycerophosphate, human transforming growth factor-β1 (TGF-b), hen egg-white lysozyme, human insulin, 3-isobutyl-1-methylxanthine (IBMX), alizarin red, oil red O, toluidine blue, 4’,6-diamidino-2-phenylindole (DAPI), fluorescamine, FBB alkaline solution, and leukocyte alkaline phosphatase kits were purchased from Sigma-Aldrich (St. Louis, Missouri, USA). Dulbecco’s modified Eagle’s medium (DMEM), glutamine, sodium pyruvate, streptomycin, penicillin, phosphate-buffered solution (PBS, pH 7.4), and 3-(4,5-dimethylthiazol-2-yl)-2,5-diphenyl tetrazolium bromide (MTT) were purchased from PanEco (Moscow, Russia). Fetal bovine serum (FBS) was from PAA Laboratories GmbH (Pasching, Austria), 2-mercaptoethanol was from Loba Feinchemie AG (Fischamend, Austria), dimethyl sulfoxide (DMSO) was from Helicon (Moscow, Russia), and calcein AM was from Abcam (Cambridge, UK). RNeasy Mini Kit was from Qiagen (Hilden, Germany), while MMLV RT Kit and qPCRmix-HS SYBR+LowROX Kit were purchased from Evrogen (Moscow, Russia). 

Chitosan (MW 80 kDa, DD 0.89) was prepared from crab chitin (Xiamen Fine Chemical, Jinan, China) by the authors through solid-state mechanochemical synthesis as described earlier [39,40]. The copolymers were synthetized by solid-state mechanochemical treatment of chitosan with semi-crystalline oligo (L,L-lactide) (MW 5 kDa) or amorphous oligo (L,D-lactide) (MW 5 kDa) as described by us earlier [35].

### 2.2. Preparation of Films and Macroporous Hydrogel Samples 

Two types of matrices for cell cultivation were obtained, namely films (2D) and macroporous hydrogels (3D). The film samples were fabricated from Chit or Chit-oligolactide copolymers (2%, *w*/*v*, in 2% acetic acid) solutions by casting using polystyrene Petri dishes (9 cm in diameter). The films were dried in a dust-free chamber at room temperature for 48 h. The sizes of the films were equal to the well diameters of the plates. To study cell morphology, we used the films with the diameter of 6.4 mm (in a 96-well plate), while, the films (10.6 mm in diameter) were placed on the bottom of the 48-well plate to assess an expression of differentiation genes by qRT-PCR. The thickness of the films was approximately 100 µm.

Macroporous hydrogel samples were prepared by lyophilization of 2% (*w*/*v*) polymer or copolymer solutions in 4% acetic acid. The hydrogel samples (4 mm in diameter, 2 mm in thickness) were prepared from chitosan, chitosan-g-oligo(L,L-lactide), or chitosan-g-oligo(L,D-lactide) copolymers, washed with milliQ water and then freeze-dried.

To convert the films and the hydrogels into a water-insoluble form, they were heated at 150 °C for 5 h [36]. The obtained samples (the films or the hydrogels) were marked as Chit, Chit-LL, or Chit-LD. The film and hydrogel samples from Chit were used as control samples.

### 2.3. Characterization of the Macroporous Hydrogels

#### 2.3.1. Study of the Macroporous Hydrogel Morphology

The structure of the swollen macroporous hydrogel samples was evaluated by confocal laser scanning microscopy using a Nikon TE-2000 inverted microscope equipped with an EZ-C1 confocal laser (Nikon, Tokyo, Japan). For this purpose, the macroporous hydrogel samples were swollen in PBS (pH 7.4) and stained with fluorescamine (0.3 μg/mL in acetone) to provide amino-specific staining. The excitation wavelength was 408 nm and fluorescence signals were collected at 515±30 nm. Image analysis software (ImageJ, National Institutes of Health, USA) was used for 3D reconstruction of the macroporous hydrogel structure and to determine the mean pore size. An effective pore diameter (d) was calculated as follows:d = (L × S)^1/2^,
where L is a pore long axis length and S is a pore short axis length. The mean pore size was determined by randomly measuring at least 100 pores for each macroporous hydrogel sample.

#### 2.3.2. Macroporous Hydrogel Swelling Degree Measurements

The swelling degree of the obtained macroporous hydrogel samples was studied using a gravimetric method. The macroporous hydrogel samples were pre-weighted and incubated in DMEM for 24 h. Then, the swollen macroporous hydrogels were again weighted. The equilibrium swelling ratio (Se) of the hydrogels was calculated as follows:Se = (M_w_ − M_d_)/M_d_,
where M_d_ is the weight of the dry macroporous hydrogel sample and M_w_ is the weight of the swollen macroporous hydrogel sample.

#### 2.3.3. Study of Macroporous Hydrogel Degradation In Vitro

The degradation of the macroporous hydrogels was investigated gravimetrically by measuring the weight loss of the samples. For this purpose, the hydrogel samples were pre-weighed, sterilized, and incubated in DMEM or in lysozyme solution (2 mg/mL) in PBS (pH 7.4). The solutions were refreshed once a week. After 7, 14, 21, and 31 days of incubation the macroporous hydrogel samples were washed three times with milliQ water and dried at 50 °C to the constant weight. The weight loss (W_L_) was calculated as follows:W_L_ (%) = (W_0_ − W_1_)/W_0_ × 100%,
where W_0_ and W_1_ are the weights of the samples before and after degradation, respectively.

### 2.4. In Vitro Evaluation

#### 2.4.1. Cell Cultivation

In the current study, mouse fibroblasts (L929), obtained from the Collection of Vertebrate Cell Cultures (Institute of Cytology of the Russian Academy of Sciences), and human mesenchymal stromal cells were used. Human mesenchymal stromal cells were isolated from adipose tissue as described earlier [41], and the first 6 passages were used. The L929 cells and MSCs were cultured in DMEM supplemented with 10% FBS and containing 2 mM L-glutamine, 1 mM sodium pyruvate, 50 μM 2-mercaptoethanol, 100 μg/mL streptomycin, and 100 U/mL penicillin. The cells were cultured in a 5% CO_2_ humidified atmosphere at 37 °C.

#### 2.4.2. Characterization of Human Adipose Tissue Derived MSCs

In order to verify the tri-lineage differentiation of the MSCs, the cells of the passage 2 were cultured in a 6-well plate up to 80% confluence, and then a culture medium was replaced with an induction medium. The osteogenic induction medium (OS) was used for MSCs differentiation into osteoblasts, it contained 100 nM dexamethasone, 10 mM b-glycerol phosphate, and 0.05 mM ascorbate-2-phosphate. The addition of 1 μg/mL insulin, 0.5 mM IBMX, and 0.5 μM dexamethasone to the culture medium stimulated the directed adipogenic differentiation. For chondrogenic differentiation of MSCs, induction medium which contained 2-phosphate-ascorbic acid (500 μg/mL), sodium pyruvate (100 μg/mL), dexamethasone (40 ng/mL), and TGF-b (10 ng/mL) was used. In the control wells, MSCs were cultivated in the culture medium without inducers. The medium was refreshed every 96 h.

Osteogenesis, adipogenesis, and chondrogenesis were studied after 14 days of cell cultivation using a phase contrast microscope (Primo Vert Carl Zeiss, Oberkochen, Germany) after staining with alizarin red [42], oil red O [43], and toluidine blue [44], respectively.

#### 2.4.3. Sterilization of the Matrix Samples

All film and macroporous hydrogel samples were sterilized by incubation in a 70% alcohol solution for 1 h. Then they were washed three times with a PBS solution (pH 7.4) and incubated in the culture medium for 24 h.

#### 2.4.4. Cytotoxicity Study

The in vitro cytotoxicity of the macroporous hydrogel samples was studied in an extract test using L929 mouse fibroblasts as model cells. For this purpose, the sterilized macroporous hydrogel samples were incubated in DMEM supplemented with 10% FBS (25 mg of sample per 1 mL) at 37 °C. After 24 h, the obtained extracts were collected. The L929 cells were seeded in a 96-well plate (10^4^ cells/well) and the plates were transferred to the CO_2_ incubator. Then the medium in each well was replaced with 100 μL of the extract in 24 h. As a control, the cells cultivated in the medium without the extract were used. Cell viability was measured by the MTT-assay. After 24 h, the extract in each well was replaced with 100 μL of an MTT solution (0.5 mg/mL in DMEM), and the plates were incubated at 37 °C for 1 h. Then formazan crystals formed in the living cells were dissolved with DMSO (100 μL per well), and an optical density was measured at 540/690 nm using a Multiscan plate reader (Flow Laboratories, McLean, VA, USA). Relative cell viability (V) was calculated as follows:V = (OD_t_/OD_c_) × 100,
where OD_t_ is an optical density in testing wells, while OD_c_ is an optical density in control wells. Results were expressed as the mean ± standard deviation for three replicates.

#### 2.4.5. Cell Cultivation in the Macroporous Hydrogels

For cell cultivation, all hydrogel samples were prepared as described earlier (see Section 2.4.3). The cells were seeded by dropping cell suspension to the hydrogel samples, which were previously transferred to 96-well plates. Fibroblasts L929 and MSCs were seeded at a density of 2 × 10^4^ and 4 × 10^4^ cells/sample, respectively.

The cell morphology was evaluated by CLSM after cultivation for 3 days. The hydrogel samples with cells were incubated with solutions of calcein AM dye (1 μg/mL) and DAPI (10 μg/mL) in the culture medium at 37 °C for 30 min. Then supernatants were replaced with fresh medium, and the samples were studied. The excitation wavelength was 488 nm for calcein AM and 358 nm for DAPI. Fluorescence signals were collected at 500–530 nm for calcein AM and 460 nm for DAPI.

Cell viability was evaluated by MTT-assay on days 3, 7, and 10 in the case of the L929 cells, and on days 7, 10, and 14 in the case of MSCs. For this purpose, the macroporous hydrogel samples with the cells were transferred to fresh wells, 100 μL of the MTT solution in the culture medium (0.5 mg/mL) was added to each well, and then the plate was incubated for 1.5 h at 37 °C. The obtained formazan crystals were dissolved in DMSO (200 μL/well), and 100 μL aliquots were taken to measure optical density (540/690 nm) at the Multiscan reader. The Chit macroporous hydrogel sample and a monolayer cell culture were taken as negative and positive controls, respectively.

To estimate an impact of the hydrogels to the results of MTT-assay, in particular cell numbers, calibration curves in the presence of the appropriate samples were plotted. For this purpose, the macroporous hydrogel samples were prepared as described above (see Section 2.4.3) and incubated in the culture medium in the CO_2_ incubator for 7 days. Then the L929 mouse fibroblasts or mesenchymal stromal cells were plated (96-well plate, 5–20 × 10^4^ cells/well), and the plates were transferred to the CO_2_ incubator for 3 h for cell attachment. Finally, the pre-incubated blank macroporous hydrogel samples (without cells) were added to the previously attached cells, and the MTT-assay was carried out. Monolayer culture (without the hydrogel sample) was used as a control. For each sample, a calibration curve was plotted: the optical density of the cells in the presence of the hydrogel sample (abscissa X) against the number of cells (ordinate Y). The number of cells in each macroporous hydrogel sample was calculated using an appropriate curve.

#### 2.4.6. Cultivation of Cells on the Films

The film samples were prepared as described earlier (see Section 2.4.3). The L929 mouse fibroblasts and MSCs were seeded by dropping cell suspensions directly to the film samples (1 × 10^4^ cells/sample) and observed by light microscopy on day 4.

To study cell differentiation, MSCs (passage 3–5) were seeded on the film samples previously placed into a 96-well plate (1 × 10^4^ cells/well) or 48-well plate (4x10^4^ cells/well) and cultured in DMEM (10% FBS) for 7 days until 80% confluence was reached. To induce differentiation, the cells were cultivated in OS medium for 7-14 days. The cells cultured in DMEM (10% FBS) were used as a control.

To estimate alkaline phosphatase activity, mesenchymal stromal cells cultured on the film samples in a 96-well plate were used. For this purpose, MSCs were washed twice with PBS (pH 7.4) and then fixed by incubation in a 4% paraformaldehyde solution for 20 min. Then, the cells were washed 3 times with PBS (pH 7.4) and with milliQ (200 μL/well). A leukocyte alkaline phosphatase kit was used for a qualitative assessment of alkaline phosphatase activity following the manufacturer’s instructions. Briefly, 100 μL of the phosphatase was added to each well and incubated for 15 min. Then the cells were rinsed with milliQ (200 μL/well) until the solution was clear. Cell morphology and distribution of differentiated cells on the film samples were studied using light microscopy.

To evaluate MSCs differentiation by qRT-PCR, the cells were cultured on the films in a 48-well plate. After cultivation for 7 and 14 days, the film samples were washed with PBS (pH 7.4), and then RNA was isolated using RNeasy Mini Kit, according to the manufacturer’s instruction. Reverse transcription was carried out using MMLV RT Kit, following the manufacturer’s instruction. qRT-PCR was conducted using qPCRmix-HS SYBR+LowROX Kit and following primers: *Runx*2 (FV: CGGAATGCCTCTGCTGTTAT; RV: TGTGAAGACGGTTATGGTCAAG), *ALPL* (FV: TGGAGTATGAGAGTGACGAGAA; RV: GTTCCAGATGAAGTGGGAGTG), *SPP*1 (FV: CCGAGGTGATAGTGTGGTTTATG; RV: CTTTCCATGTGTGAGGTGATGT), *GAPDH* (FV: TCGACAGTCAGCCGCATCTTCTTT; RV: ACCAAATCCGTTGACTCCGACCTT).

### 2.5. Statistics

All values were expressed as the mean ± standard deviation of at least three parallel replicates and compared using the paired samples *t*-test and IBM SPSS Statistics for Windows, version 22. The values of *p* < 0.05 were considered as significant.

## 3. Results and Discussions

In the current study, two types of matrices were obtained, namely the 2D films, where the cells were localized and grown on the film surface, and 3D hydrogels providing cell growth both on a hydrogel surface and within hydrogel macropores.

### 3.1. Study of the Macroporous Hydrogel Structures

As is well known, a structure of macroporous hydrogels is of great importance, since it should provide an optimal specific surface area for cell attachment and growth, as well as a nutrient diffusion, gas exchange, and neovascularization. In this study, the macroporous hydrogel structures in a swollen state were studied by confocal laser microscopy. It was shown that the structures of the swollen macroporous hydrogel samples were arranged as developed systems of interconnected pores (Figure 1).

The mean pore sizes of all macroporous hydrogels were calculated from CLSM images and were found to vary in a range of 50–400 µm (Figure 2). Thus, the mean pore sizes for Chit, Chit-LL, and Chit-LD samples were found to be 150 ± 5 μm, 147 ± 15 μm, and 148 ± 8 μm, respectively. As for the copolymer hydrogel samples, a formation of both primary and secondary pores within hydrogel walls was revealed. This effect as well as wider pore size distribution was more pronounced for the Chit-LL sample due to the lower cross-linking efficiency of this copolymer compared to the Chit-LD sample. It should be noted that the mean pore sizes in all samples were within a size range suitable for animal cell growth and proliferation [45].

### 3.2. Study of Macroporous Hydrogel Swelling Behavior

An extracellular matrix in tissues consists of proteins containing both hydrophilic and hydrophobic amino acids, providing protein–protein interactions as well as a regulation of protein folding, bioactivity, etc. A study of the swelling behavior of hydrogel matrices is of great value, since it allows for an estimation of their hydrophilicity. Optimization of matrix swelling behavior is also of great importance, since an enhanced swelling of scaffolds can influence mechanical properties, and lead to loosening and displacement of the implant [46,47]. Moreover, earlier, it was shown that the scaffold swelling behavior can affect cell adhesion [48,49]. Thus, possible adjustment of the hydrophilic –hydrophobic balance is a promising approach to improve scaffold bioadhesive properties. Recently, we have shown that copolymer hydrogels’ swelling capacity in milliQ water was dependent on hydrogel composition [36]. Here, we studied the hydrogel swelling behavior in DMEM, since it could depend on the solution’s ionic strength. Therefore, it is important to study the matrix behavior in the culture medium before cell cultivation in vitro. In the current study, swelling of the Chit macroporous hydrogel in DMEM was found to be 27.8 ± 1.8 mL/g (Figure 3). In case of the Chit-LL hydrogel sample, the swelling degree slightly increased up to 31.1 ± 3.5 mL/g. Earlier, Luckachan et al. [28] reported that swelling capacity values of matrices from graft copolymers of chitosan with oligo-L,L-lactide were higher than those of Chit matrix, which is in a good agreement with our results. It is likely that grafting oligo-L,L-lactide onto chitosan separates the main Chit chains and sharply reduces a number of hydrogen bonds and crystallinity. Additionally, it could result in an increase in its affinity for water. Similar results were reported for copolymers of chitosan with poly(ethylene glycol) by Gorochovceva et al. [50]. Thus, swelling of the graft copolymers in water can increase, despite the hydrophobicity of the L,L-lactide side chains. As seen in Figure 3, the swelling degree of the Chit-LD hydrogels was lower (23.6 ± 1.1 mL/g) than that of the Chit hydrogel (27.8 ± 1.8 mL/g). This result is also in a good agreement with our previous result for the hydrogel from the Chit-LD copolymer, which had a shorter lactide chain, in particular, the degree of polymerization of grafted chains was 4.1 instead of ≤70 in this study [26]. Therefore, hydrophobic oligolactide Chit-LD imparted amphiphilic properties to the matrices.

### 3.3. Study of the Macroporous Hydrogel Degradation

The biodegradation kinetics should be in a good agreement with cell proliferation and scaffold replacement with novel repaired tissues. Thus, with regeneration of load-bearing cardiovascular tissues, slow degradation is of great importance in order to maintain collagen orientation [51]. On the other hand, scaffolds with a rather high biodegradation rate are used for cartilage regeneration, since they can replace damaged tissues (defects) rather fast [52]. Hydrogel biodegradation has to provide some room for cell migration and a deposition of ECM molecules, which can interact with cells inducing specific signaling pathways for cell binding, migration, and differentiation. The production and accumulation of ECM were shown to enhance with increasing cell proliferation observed for MSCs in matrices with an enhanced degradation ability [53]. However, if the matrices’ biodegradation is too rapid, one can observe a decrease in a number of viable cells due to early loss of mechanical integrity [53]. Thus, this parameter is of great importance to provide desirable cell proliferation.

The weight loss of the macroporous hydrogels from Chit or its copolymers with oligolactides was measured after their incubation for 31 days in DMEM or in lysozyme solution (2 mg/mL of PBS, pH 7.4) at 37 °C (Figure 4). It can be seen that the degradation of all hydrogel samples was faster in the lysozyme solution than in the culture medium. After 21 days, the weight loss values of the Chit and Chit-LD samples reached 35% and 27%, respectively. The maximum resorption degrees, namely 47% and 58% in DMEM and the lysozyme solution, respectively, were found in the Chit-LL hydrogel sample on day 31. It could be explained by lower efficacy of thermal treatment of the Chit-LL sample compared to that of other hydrogels [36]. It should be noted that the degradation degree values correlated well with the results on the swelling behavior of the samples (see Section 3.2). Moreover, a similar increase of hydrogel biodegradation with a swelling enhancement was shown previously [28,32].

### 3.4. In Vitro Cytotoxicity Study

As is well known, in vitro cytotoxicity evaluation is the first step in the development of biomaterials for tissue engineering. Here, we studied a cytotoxic effect of the extracts obtained after incubation of the hydrogel samples in DMEM (10% FBS) for 24 h. The L929 mouse fibroblasts were used as model cells, while the number of viable cells was estimated by MTT-test. As seen in Figure 5a, all extracts were found to be non-toxic for the cells, while the numbers of viable cells did not differ significantly for all hydrogel samples.

The morphology of L929 cells after their incubation in the extracts was observed by light microscopy. As seen in Figure 5b, the incubation with the extracts did not lead to any change in the cell morphology. Thus, the hydrogels were non-toxic, and therefore they were used for further experiments in order to study cell growth and proliferation at long-term cultivation in vitro.

### 3.5. Cell Cultivation on the Films and in Chitosan-Based Macroporous Hydrogels In Vitro

To study cell adhesion and morphology as well as cell growth and proliferation in/on the matrices, two types of cells, namely L929 mouse fibroblasts and MSCs were used. As mentioned above, the microporous films were here considered as 2D matrices, where the cells were localized and grew only on the film surface, while the hydrogels, due to their macroporous structure, provided 3D cell growth both on the surface and within the pores. Differentiation of MSCs was evaluated using the Chit-based films, but previously cell phenotype and multipotency of these cells were estimated.

#### 3.5.1. Characterization of MSCs Phenotype

In this study, human adipose-derived MSCs were used after isolation and expansion (at passage 2) with fusiform morphology and homogeneous distribution. The tri-lineage differentiation experiments showed that MSCs were able to differentiate into osteoblasts, adipocytes, and chondrocytes (see Appendix A). Thus, both the phenotype and the multipotency of MSCs were confirmed.

#### 3.5.2. Morphology of Cells Cultured on Chitosan-Based Films

As is well known, TE matrices should provide an appropriate surface chemistry and topography to support cell adhesion, morphology, and proliferation or differentiation. To study cell morphology, L929 cells and MSCs were cultivated on copolymer-based films and observed by optical microscopy for 4 days. As seen in Figure 6, the surface properties of the films indeed affected cell morphology. First, the L929 cells were found to be well attached to all films. Secondly, in the case of the film samples from Chit-LL (Figure 6b) and Chit-LD (Figure 6c), the L929 cells were characterized by a spindle-like shape, while on the Chit film (Figure 6a) a lot of flattened cells that were round in shape were also observed.

As for MSCs, the cellular microenvironment is known to have a profound effect on the biology of the stromal cells. For example, stromal cells’ differentiation is highly regulated by their niche through both intrinsic and extrinsic cues. It should be noted that MSCs can spontaneously form spheroids on chitosan films, as described earlier [54]. However, in our study, most of the cells had a typical spindle-like shape on all film samples (Figure 6e–g). The differences in cell behavior could be associated with chitosan and its characteristics, such as molecular weight and deacetylation degree, a source of MSCs, as well as with an initial cell density [55].

#### 3.5.3. Morphology of Cells Cultivated in Chitosan-Based Macroporous Hydrogels

The morphology and distribution of the L929 cells and MSCs within the macroporous hydrogel were studied by CLMS after cell cultivation for 3 days (Figure 7). The cells were stained with a vital calcein AM dye (in green), while the hydrogels were visualized by non-specifically absorbed DAPI (in blue). The L929 cells were viable and rather evenly distributed within the hydrogel structure after cultivation for 3 days.

Moreover, the hydrogel composition was found to affect cell morphology. For instance, in case of the Chit hydrogel, L929 mouse fibroblasts had a typical fusiform morphology (Figure 7a). They were well spread and evenly distributed within the hydrogel. As for the Chit-LL sample, some cells were found to be adherent but non-spread (round in shape). In case of the Chit-LD hydrogel, most of the cells were spherical in shape but also rather evenly filled the macropores. This could be related to the differences in the hydrophilic–hydrophobic balance, which occurred in the copolymer-based matrices compared to the Chit hydrogel. After cultivation for 3 days in the hydrogels, MSCs were found to have a fusiform morphology in all hydrogel samples (Figure 7c–e). However, several round-shaped cells were also observed in the Chit-LL and Chit-LD samples.

Thus, the structure and surface chemistry of the hydrogels provided cell migration within the interconnected hydrogel macropores and cell attachment to the pore surface. Most of the cells were shown to be spread over the pore surface and had the typical morphology.

#### 3.5.4. Cell Growth and Proliferation in the Macroporous Hydrogels

The L929 fibroblasts and MSCs were cultured in the hydrogel samples for 10 and 14 days, respectively. To quantify numbers of alive cells in the samples, an MTT-test was used. As seen in Figure 8, the cells of both types proliferated in all macroporous hydrogel samples. Moreover, the number of viable cells depends on the hydrogel composition. Thus, the maximum growth of mouse fibroblasts L929 was revealed for the Chit hydrogel samples. The growth of fibroblasts in the Chit-LL hydrogels was comparable to the result obtained for the Chit hydrogel (Figure 8a). The smallest number of proliferating L929 cells was observed in case of the Chit-LD sample. In case of MSCs, the numbers of viable MSCs on days 10 and 14 of cultivation in copolymer macroporous hydrogels were comparable to the control (bottom of the plate well), while the largest number of MSCs was observed in the case of the Chit sample (Figure 8b).

### 3.6. Cell Differentiation

As is well known, MSCs can differentiate into osteoblasts (bone cells), chondrocytes (cartilage cells), and adipocytes (fat cells). In this study, we carried out an induced differentiation of MSCs into osteoblasts. Many factors are known to influence this process. For instance, as mentioned above, topography and surface chemistry should be taken into consideration. Here, we evaluated an influence of copolymer-based films on early stages of MSCs osteoinduction. To confirm the osteogenic differentiation of MSCs, alkaline phosphatase activity was detected histochemically. This approach allowed us to estimate the osteodifferentiation qualitatively. As seen in Figure 9, the MSCs demonstrated positive staining on various substrates, which confirmed their osteocommitment. The cells on the films, unlike in a control (a bottom of a 96-well plate), were not evenly distributed but formed clusters, so-called nodules. The process was more pronounced on the Chit-LL and Chit-LD films and could reflect enhanced osteogenic potential of the cells.

The MSCs were cultured on the copolymer-based films in OS medium. To study differential expression of genes involved in osteodifferentiation, mRNA was isolated using RNeasy Mini Kit for RT-PCR after 7 and 14 days of cultivation. The results are shown in Figure 10. The osteodifferentiation-related genes, such as *ALPL* [56], *Runx*2, and *SPP*1 [57] were upregulated after osteoinduction in all samples. *ALPL* is a transient early marker of the osteodifferentiation in MSCs, and activity is known to peak at the end of the proliferative stage and before matrix maturation [58]. *Runx*2 is an master osteogenic transcription factor involved in MSCs differentiation into osteoblasts [59]. *SPP*1 is an acidic phosphoprotein associated with cell attachment, proliferation, and extracellular matrix mineralization in bones. In particular, *SPP*1 is involved in the attachment of osteoblasts to the bone mineral matrix. It is highly expressed at the stage of the osteoblast proliferation as a marker of the osteoblast differentiation [60].

After 7 days of induction, a pronounced *ALPL* elevation was observed for the Chit-LL and Chit-LD samples (Figure 10). These results were in a good agreement with histochemical data (see Figure 9). On the 14th day of osteoinduction, the *ALPL* transcription was additionally upregulated compared to that on day 7, and it did not differ in all osteo-induced samples. The expression levels of *Runx*2 and *SPP*1 on day 14 were higher than those on day 7. On day 14, the maximal expression levels of all three bone-specific markers were detected on the Chit-LD films (* *p* < 0.05), which could be related to the change of physicochemical properties of the copolymer-based films compared to the Chit film. Thus, osteodifferentiation was shown to be in function of the film composition.

## 4. Conclusions

Biodegradable matrices in the form of the 2D films and the 3D macroporous hydrogels based on copolymers of chitosan with oligo(L,L-/L,D-lactides) were obtained and characterized in terms of their physical–chemical properties and their ability to support cell proliferation or differentiation in vitro. The macroporous hydrogels were fabricated using a freeze-drying technique. For the first time, swelling and degradation of these copolymer-based macroporous hydrogels were studied and shown to depend on the hydrogel composition. Thus, as compared to the Chit sample, for the matrices from copolymers of Chit with oligo(L,L-lactide) the swelling and degradation degrees increased by 12% and 34%, respectively, while for the Chit-LD hydrogel sample these parameters decreased by 15% and 23%, respectively. Cytotoxicity of the macroporous hydrogels was studied using mouse fibroblasts (L929) as model cells. All samples were shown to be non-toxic. Both copolymer hydrogels were demonstrated to support good cell adhesion, growth, and proliferation after long-term cultivation of mouse fibroblasts L929 and mesenchymal stromal cells, which was confirmed by confocal microscopy and MTT-assay. Osteogenic differentiation of MSCs was evaluated using both chitosan and copolymer-based films. After cultivation of MSCs for 14 days, maximal expression levels of osteogenesis markers (*ALPL*, *Runx*2, *SPP*1) were revealed by qRT-PCR for the Chit-LD films. Thus, the matrices based on chitosan-g-oligo (L,L-/L,D-lactide) copolymers are promising for tissue engineering, while the osteodifferentiation was shown to be in function of the film composition.

## Figures and Tables

**Figure 1 polymers-15-00907-f001:**
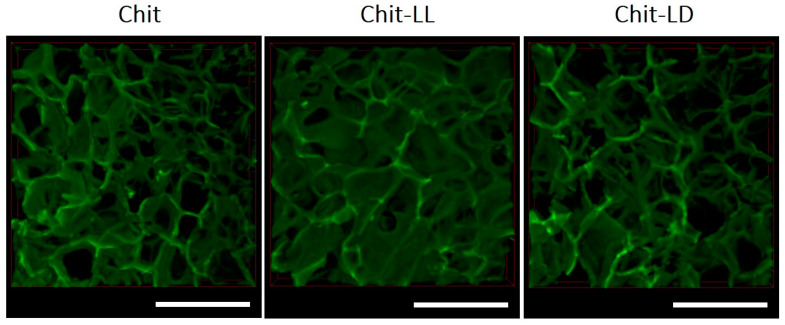
CLSM images of the 3D macroporous hydrogel samples produced from chitosan (Chit) and two copolymers: chitosan-g-oligo(L,L-lactide) (Chit-LL) and chitosan-g-oligo(L,D-lactide) (Chit-LD). The scale bar is 500 µm.

**Figure 2 polymers-15-00907-f002:**
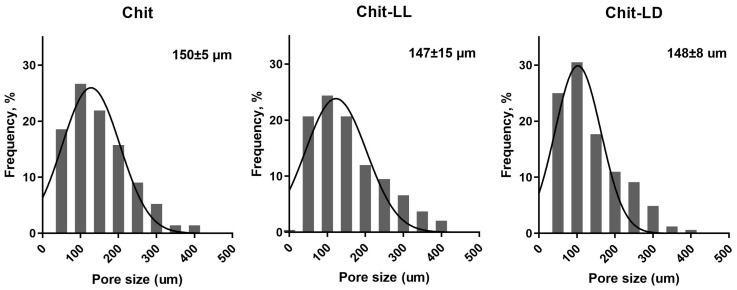
Pore size distribution in the macroporous hydrogel samples from chitosan (Chit) and copolymers: chitosan-g-oligo(L,L-lactide) (Chit-LL) and chitosan-g-oligo(L,D-lactide) (Chit-LD).

**Figure 3 polymers-15-00907-f003:**
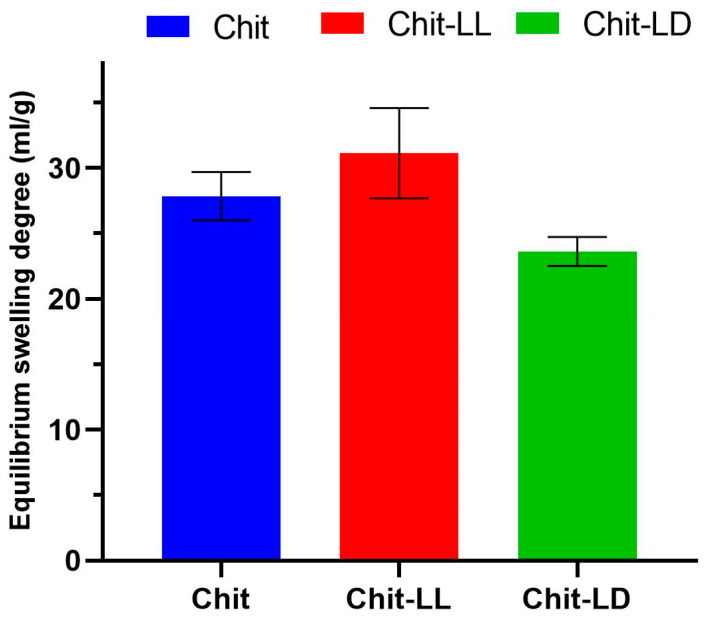
Equilibrium swelling degrees of the macroporous hydrogels from chitosan (Chit), chitosan-g-oligo(L,L-lactide) copolymer (Chit-LL); and chitosan-g-oligo(L,D-lactide) copolymer (Chit-LD) after incubation in DMEM for 24 h. Data are expressed as the mean ± SD. All experiments were performed in triplicate.

**Figure 4 polymers-15-00907-f004:**
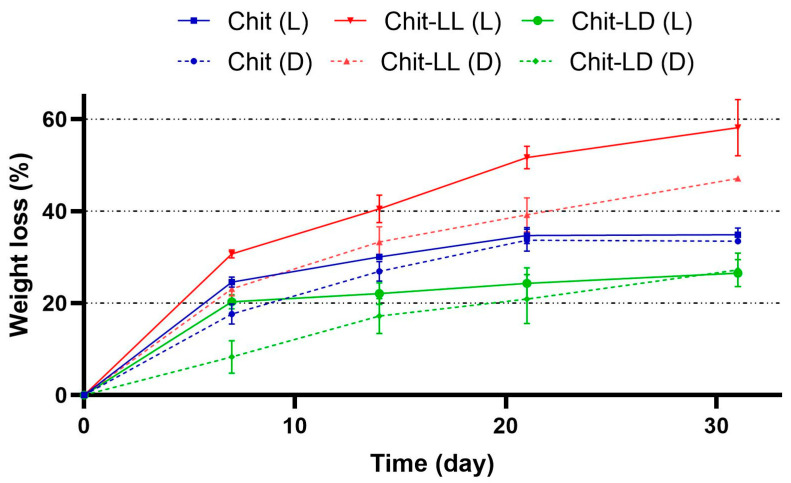
Degradation kinetics of the macroporous hydrogel samples from chitosan (Chit), chitosan-g-oligo(L,L-lactide) (Chit-LL), and chitosan-g-oligo(L,D-lactide) (Chit-LD) copolymers after their incubation in DMEM (D) or in lysozyme (L) (2 mg/mL) for 31 days at 37 °C. Data are expressed as the mean ± SD. All experiments were performed in triplicate.

**Figure 5 polymers-15-00907-f005:**
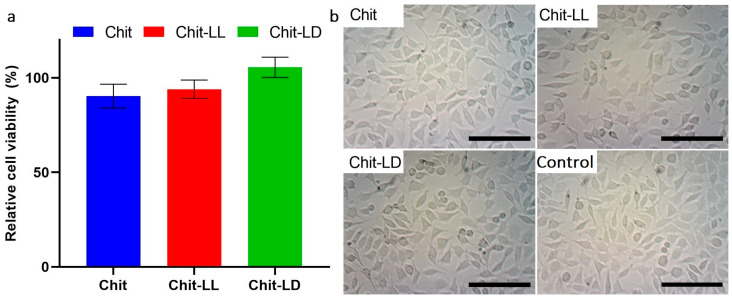
Cell viability after incubation of the L929 mouse fibroblasts in the extracts for 24 h (**a**) and cell morphology (**b**). The results of the MTT-assay. The cells cultivated in DMEM (10% FBS) were taken as a control (100%). Optical microscopy. The scale bar is 100 µm (magnification ×100). Data are expressed as the mean ± SD. All experiments were performed in triplicate.

**Figure 6 polymers-15-00907-f006:**
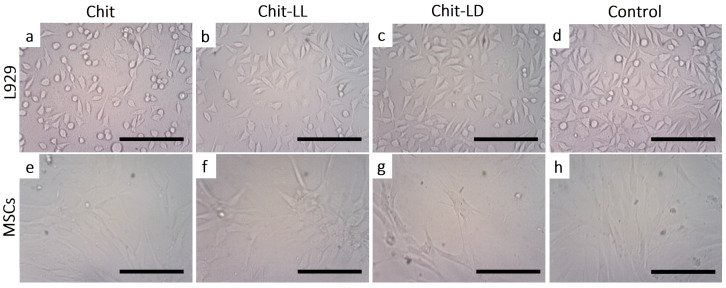
Micrographs of L929 mouse fibroblasts (**a**–**d**) and mesenchymal stromal cells (**e**–**h**) on the chitosan (Chit), chitosan-g-oligo(L, L-lactide) (Chit-LL), and chitosan-g-oligo(L, D-lactide) (Chit-LD) films after cultivation for 4 days. Monolayer cell culture (**d**,**h**) were taken as controls. Optical microscopy. The scale bar is 100 µm (magnification ×100).

**Figure 7 polymers-15-00907-f007:**
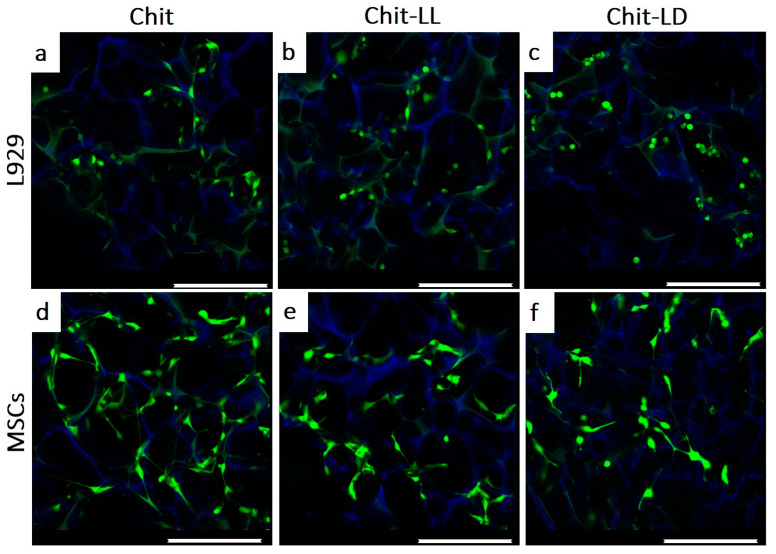
CLSM images of the L929 mouse fibroblasts (**a**–**c**) and MSCs (**d**–**f**) after cultivation in the macroporous hydrogels from chitosan (Chit), chitosan-g-oligo(L,L-lactide) (Chit-LL), and chitosan-g-oligo(L,D-lactide) (Chit-LD) hydrogels for 3 days. The alive cells (in green) and the hydrogels (in blue) were stained with calcein AM and DAPI, respectively. The scale bar is 250 µm.

**Figure 8 polymers-15-00907-f008:**
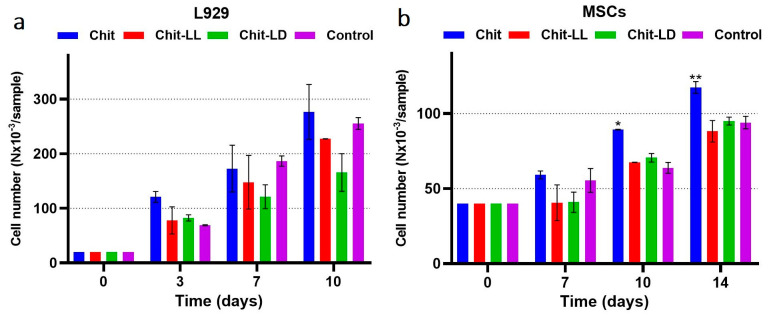
Cell growth of the L929 mouse fibroblasts for 10 days (**a**) and MSCs for 14 days (**b**) in the chitosan (Chit), chitosan-g-oligo(L,L-lactide) (Chit-LL), and chitosan-g-oligo(L,D-lactide) (Chit-LD) hydrogels. Results of MTT-test. Monolayer cell culture was taken as a control (100%). Data are expressed as the mean ± SD. * *p* < 0.05, ** *p* < 0.01, versus control. All experiments were performed in triplicate.

**Figure 9 polymers-15-00907-f009:**
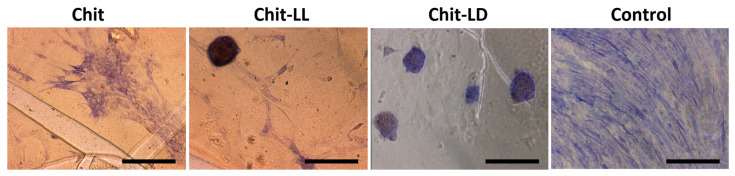
Induced osteodifferentiation of MSCs on the films from chitosan (Chit), chitosan-g-oligo(L,L-lactide) (Chit-LL), and chitosan-g-oligo(L,D-lactide) (Chit-LD). Evaluation of alkaline phosphatase activity. Representative images of the histochemical reaction. The scale bar is 100 µm (magnification ×100). Monolayer cell culture was taken as a control. MSCs were cultured in osteogenic medium (OS) for 7 days.

**Figure 10 polymers-15-00907-f010:**
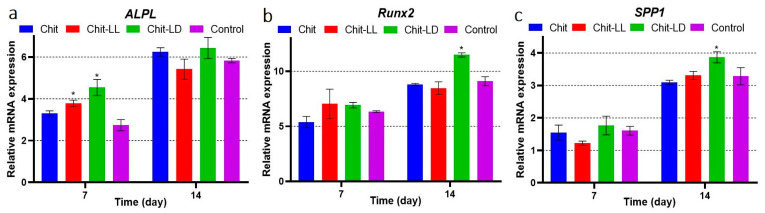
Differential expression of genes involved in osteodifferentiation. MSCs were cultured on the chitosan (Chit), chitosan-g-oligo(L,L-lactide) (Chit-LL), and chitosan-g-oligo(L,D-lactide) (Chit-LD) films. Monolayer cell culture was taken as a control. MSCs were cultured in osteogenic medium (OS) for 14 days. The data of the qRT-PCR analysis of bone specific markers *ALPL*, *Runx*2, and *SPP*1. The data are presented as fold changes of transcription levels of osteo-induced MSCs vs. MSCs. Data are expressed as the mean ± SD. All experiments were performed in triplicate.

## Data Availability

Data are contained within the article.

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
