# Peer review of "Preparation and In Vitro Evaluation of Chitosan-g-Oligolactide Based Films and Macroporous Hydrogels for Tissue Engineering"

_polymers, 2023, doi:10.3390/polym15040907_

Round 1
Reviewer 1 Report
Manuscript No.: polymers-2194371-peer-review-v1
Title: Biodegradable films and macroporous hydrogels based on chitosan-g-oligolactides for tissue engineering: fabrication and in vitro evaluation
Polymers
Reviewer’s Decision: Accept after major revision
The authors of this research work describe the fabrication of chitosan-based hydrogels for tissue engineering and their in vitro evaluations. The research is significant and should be published in the Polymers. However, before it can be published, the manuscript needs to be significantly improved. As a result, I recommend accepting the manuscript after significant and satisfactory revisions. The following are the detailed comments:
1. Title: The title appears confusing; the author has developed biodegradable films and hydrogels for tissue engineering. It is, therefore, recommended to update the title to avoid confusion for the readers. I have the following suggestion
“Fabrication of chitosan-g-oligolactides baed films and hydrogels for tissue engineering: in vitro evaluation.”
2. Abstract: The abstract is a comprehensive summary of the whole research article. The abstract contains numerous grammatical and formatting errors. It is suggested to improve the grammar and English language problems. The abstract section is more introductive and contains methodology information. It is recommended to reduce the informative and methodology section and add more results outputs with specific biomedical applications. The incomplete information in the abstract may confuse the readers.
3. Introduction: The introduction section discusses more polymer materials and provides little information about tissue engineering and other aspects of tissue engineering. The introduction and the rest of the manuscript have several grammatical and formatting errors. Please improve the grammar, language, and formatting issues in the manuscript.
4. References: The manuscript lacks the literature citation of some highly interesting, most recent relevant works; thus, the references are not up to date. Too many references have been given for a research article that may question the novelty of research work that may give an impression that so many people have reported the research already. These citations will help to explain cellular behavior against fabricated films and hydrogels, characterization and other in vitro assays. In this regard, the author should refer to some of the most recent papers on hydrogel, such as
· Hasan, A., et al. (2022). Sodium alginate-f-GO composite hydrogels for tissue regeneration and antitumor applications. International Journal of Biological Macromolecules, 208, 475-485.
· Stojanović, G. M., et al. (2022). Multifunctional Arabinoxylan-functionalized-Graphene Oxide Based Composite Hydrogel for Skin Tissue Engineering. Frontiers in Bioengineering and Biotechnology, 10.
5. Materials and methods: This section is missing, and please add it; otherwise, it may confuse the readers.
· The author has used the same “degree” sign for temperature and angle. It is recommended to use the “degree” symbol accordingly throughout the manuscript.
· Be consistent with what you are writing; as line 117, write “2%” or “2 %”
· “CO2” should be like “CO2.”
6. Results and Discussions: The following issue must be taken into consideration.
a. Why not add the SEM morphology of the fabricated sample if possible!
b. The figure caption and style should be identical throughout the manuscript to keep the continuity of the work, as they are not in your manuscript.
7. Conclusions: The conclusion section is the most important summary of a research article, and it should be based on the conclusion for the conclusion. The conclusion should be based on comparing the different used formulations by comparing the best result output. But it is more introductory and explanatory than the conclusion based, and it is advised to revise the conclusion section.
8. As per the comments given for the results and description.
In summary, the reported work has significant value; however, a major and thorough improvement/correction of language, grammar, syntax, etc., is necessary to improve the quality of the paper and to make it publishable in the Polymers.
· All the abbreviations should be defined before their 1st-time use.

Author Response
We are grateful to the editor and the reviewers for the valuable advices and for the opportunity to revise and improve our manuscript. Please find bellow our revisions, answers and comments. The changes and modifications have been marked through the text of the revised manuscript.
Response to Reviewer 1 Comments
The authors of this research work describe the fabrication of chitosan-based hydrogels for tissue engineering and their in vitro evaluations. The research is significant and should be published in the Polymers. However, before it can be published, the manuscript needs to be significantly improved. As a result, I recommend accepting the manuscript after significant and satisfactory revisions. The following are the detailed comments:
- Title:The title appears confusing; the author has developed biodegradable films and hydrogels for tissue engineering. It is, therefore, recommended to update the title to avoid confusion for the readers. I have the following suggestion
“Fabrication of chitosan-g-oligolactides baed films and hydrogels for tissue engineering: in vitro evaluation.”
Response: Thank you for the comment. We changed the title of the manuscript according to your suggestion with minor changes. The word “macroporous” is of great importance, and therefore we would prefer to keep it in the title.
- Abstract: The abstract is a comprehensive summary of the whole research article. The abstract contains numerous grammatical and formatting errors. It is suggested to improve the grammar and English language problems. The abstract section is more introductive and contains methodology information. It is recommended to reduce the informative and methodology section and add more results outputswith specific biomedical applications. The incomplete information in the abstract may confuse the readers.
Response: Thank you. We have focused not only the methods but also on some results of our study. The Abstract was updated.
- Introduction:The introduction section discusses more polymer materials and provides little information about tissue engineering and other aspects of tissue engineering. The introduction and the rest of the manuscript have several grammatical and formatting errors. Please improve the grammar, language, and formatting issues in the manuscript.
Response: Some information about tissue engineering and some key requirements which should meet biomaterials for tissue engineering has been added to the Introduction. The added text is marked in red (see lines 43-58). Formatting, language and grammar errors have been also edited.
- References:The manuscript lacks the literature citation of some highly interesting, most recent relevant works; thus, the references are not up to date. Too many references have been given for a research article that may question the novelty of research work that may give an impression that so many people have reported the research already. These citations will help to explain cellular behavior against fabricated films and hydrogels, characterization and other in vitro assays. In this regard, the author should refer to some of the most recent papers on hydrogel, such as
- Hasan, A., et al. (2022). Sodium alginate-f-GO composite hydrogels for tissue regeneration and antitumor applications. International Journal of Biological Macromolecules, 208, 475-485.
- Stojanović, G. M., et al. (2022). Multifunctional Arabinoxylan-functionalized-Graphene Oxide Based Composite Hydrogel for Skin Tissue Engineering. Frontiers in Bioengineering and Biotechnology, 10.
Response: Thanks for your comment. We have introduced the references proposed by you in the text of the Introduction (see lines 55,56).
However, we would prefer to keep most of other references, which deal with the matrices based on chitosan, and in particular chitosan-oligo-L-lactide graft copolymers (Luckachan, G. E.; Pillai, C. K. S. Chitosan/Oligo L-Lactide Graft Copolymers: Effect of Hydrophobic Side Chains on the Physico-Chemical Properties and Biodegradability. Carbohydr. Polym. 2006, 64 (2), 254–266. https://doi.org/10.1016/j.carbpol.2005.11.035.; Demina, T. S.; Zaytseva-Zotova, D. S.; Akopova, T. A.; Zelenetskii, A. N.; Markvicheva, E. A. Macroporous Hydrogels Based on Chitosan Derivatives: Preparation, Characterization, and in Vitro Evaluation. J. Appl. Polym. Sci. 2017, 134 (13). https://doi.org/10.1002/app.44651.; Demina, T.; Bardakova, K.; Minaev, N.; Svidchenko, E.; Istomin, A.; Goncharuk, G.; Vladimirov, L.; Grachev, A.; Zelenetskii, A.; Timashev, P.; Akopova, T. Two-Photon-Induced Microstereolithography of Chitosan-g-Oligolactides as a Function of Their Stereochemical Composition. Polymers 2017, 9 (12), 302. https://doi.org/10.3390/polym907030.). There are no results related to effects of matrices on cell behavior, namely cell morphology, proliferation and especially differentiation in all these papers. Therefore, it is obvious that there is a novelty in our current study. On the other hand, we do need to mention these references, in order to discuss the results, which have been obtained within a frame of the current study.
- Materials and methods: This section is missing, and please add it; otherwise, it may confuse the readers.
- The author has used the same “degree” sign for temperature and angle. It is recommended to use the “degree” symbol accordingly throughout the manuscript.
- Be consistent with what you are writing; as line 117, write “2%” or “2 %”
- “CO2” should be like “CO2.”
Response: The Materials and Methods includes sections 2.1-2.5. The “degree” sign has been corrected, as well as formatting errors (%, CO2).
- Results and Discussions:The following issue must be taken into consideration.
- Why not add the SEM morphology of the fabricated sample if possible!
- The figure caption and style should be identical throughout the manuscript to keep the continuity of the work, as they are not in your manuscript.
Response:
- We agree that SEM would be of interest. However, in our study confocal microscopy was preferable over SEM, since it allowed us to evaluated the hydrogel samples in a swollen state. More over, it was possible not only to study the porous structure and to calculate mean pore sizes from color CLSM images of the hydrogel samples but in further experiments also to observe the distribution and localization of alive cells within the hydrogels. We suggest that SEM would be useful for composite hydrogels with entrapped inorganic nanoparticles, for example. But that was not our case.
- Thank you, we have made the correction.
7. Conclusions:The conclusion section is the most important summary of a research article, and it should be based on the conclusion for the conclusion. The conclusion should be based on comparing the different used formulations by comparing the best result output. But it is more introductory and explanatory than the conclusion based, and it is advised to revise the conclusion section.
Response: This paragraph was updated.
- As per the comments given for the results and description.
In summary, the reported work has significant value; however, a major and thorough improvement/correction of language, grammar, syntax, etc., is necessary to improve the quality of the paper and to make it publishable in the Polymers.
- All the abbreviations should be defined before their 1st-time use.
Response: Thank you, we have made the correction.
Reviewer 2 Report
Dear Authors,
here are my suggestions

Author Response
We are grateful to the editor and the reviewers for the valuable advices and for the opportunity to revise and improve our manuscript. Please find bellow our revisions, answers and comments. The changes and modifications have been marked through the text of the revised manuscript.
Response to Reviewer 2 Comments
In the manuscript by Tolstova and co-workers, the development of new matrices based on chitosan-goligo (L,L-/L,D-lactide) copolymers is discussed. The paper is quite interesting and can be published in Polymers.
Some comments are given below:
- The introduction to the state of the art is somewhat limited. Authors need to justify these concepts with more current and specific references in relation to tissue engineering, such as Materials 2022, 15(22), 8208; https://doi.org/10.3390/ma15228208, Int J Mol Sci. 2020 May; 21(10): 3442, doi: 10.3390/ijms21103442, and so on, should be discussed in the introduction section in order to outline the novelty of the present study.
Response: Thanks for your comments. Introduction has been revised. More over, we have added several references, including those you mentioned, in order to highlight the novelty of the present study (see lines 49 and 80). The novelty is clearly formulated in the following paragraph: «For the first time the swelling and biodegradation … etc»
- The authors need to improve the methodological explanations and current references.
Response: The Section Materials and Methods includes issues 2.1-2.5. The Section has been a bit extended and some additional references have been added.
The Section References has been also revised and extended. Several current references (approx..13 ref) have been added to the References list.
- Some important information is missing in the Materials and methods section. Please state the reasons why the 40/60 w/w composition was chosen for the production of the films.
Response: Thank you for your comment. This ratio is related to the composition used for synthesis of the copolymer samples. Both films and macroporous hydrogels were obtain from these copolymers. The optimal ratio 40/60 (w/w) was found by us earlier (Demina, T.; Bardakova, K.; Minaev, N.; Svidchenko, E.; Istomin, A.; Goncharuk, G.; Vladimirov, L.; Grachev, A.; Zelenetskii, A.; Timashev, P.; Akopova, T. Two-Photon-Induced Microstereolithography of Chitosan-g-Oligolactides as a Function of Their Stereochemical Composition. Polymers 2017, 9 (12), 302. https://doi.org/10.3390/polym9070302).
- The quality of the figures should be improved.
Response: We have improved the quality of Figures 1-3,5-8,10.
- Line 204 uses subscript for the formula.
Response: Thank you, we have corrected.
- Results support conclusions, but authors should conclude what the results show.
Response: The Conclusions have been revised based on the results obtained.
- The authors must conduct a comprehensive review of English grammar.
Response: Thank you, we have made the correction.
Reviewer 3 Report
Good job! The article is well written and informative. I do have a few comments which I think might add value:
1. Add scale bars to the Fig 1, 5b, 6 and 9. Ensure that the scalebars are legible in Fig 7.
2. Can the morphology of cells in the macroporous hydogels be repeated with Calcein AM and Ethidium Homodimer? I think it might be interesting to see how the ratio of live: dead cells.
3. Could you provide more specific details on the actual composition of the gels? Like what weight of the Chit, Chit LL or Chit LD were mixed in what volume of acetic acid? This might be helpful to others who are also trying to fabricate Chitosan based gels.
4. Please do proof reading/grammar check.
Author Response
We are grateful to the editor and the reviewers for the valuable advices and for the opportunity to revise and improve our manuscript. Please find bellow our revisions, answers and comments. The changes and modifications have been marked through the text of the revised manuscript.
Response to Reviewer 3 Comments
Good job! The article is well written and informative. I do have a few comments which I think might add value:
- Add scale bars to the Fig 1, 5b, 6 and 9. Ensure that the scalebars are legible in Fig 7.
Response: Thank you for the comment. For optical microscopy, magnifications were indicated in figure legends (see Figs. 5b, 6 and 9). As for confocal laser microscopy images, appropriate scale bars were added (see Figs. 1 and 7). The scale bars in Fig. 7 have been updated.
- Can the morphology of cells in the macroporous hydrogels be repeated with Calcein AM and Ethidium Homodimer? I think it might be interesting to see how the ratio of live: dead cells.
Response: Thank you for the comment. Indeed, it would be of great interest to estimate living/dead cells ratio in the hydrogel. However, here we were mostly focused on living cells, namely cell growth and proliferation. Therefore a vital dye Calcein AM was used. A similar approach is used by many other researches, see for example following papers [Hassani, A. et al (2022). Collagen and nano-hydroxyapatite interactions in alginate-based microcapsule provide an appropriate osteogenic microenvironment for modular bone tissue formation. Carbohydrate Polymers, 277, 118807; Neufurth, M., et al (2017). 3D printing of hybrid biomaterials for bone tissue engineering: Calcium-polyphosphate microparticles encapsulated by polycaprolactone. Acta Biomaterialia, 64, 377-388]. Unfortunately, we can not repeat the experiments with Calcein AM and Ethidium Homodimer, since this rather big and time-consuming job. But we can do it in our future study. Thanks for your advice.
- Could you provide more specific details on the actual composition of the gels? Like what weight of the Chit, Chit LL or Chit LD were mixed in what volume of acetic acid? This might be helpful to others who are also trying to fabricate Chitosan based gels.
Response: Thank you for the question. We fabricated the films and macroporous hydrogels from non-modified chitosan (Chit) and its graft-copolymers with oligolactide (Chit LL and Chit LD). All these samples were obtained from the 2% (w/v) polymer solutions in acidic aqueous solutions. Usually, we prepare 100 mL of each stock solution (2 g of Chit or Chit LL or Chit LD per 100 mL of acetic acid) and then use these solutions to fabricate the samples. The protocol in detail was reported earlier (Demina, T.; Bardakova, K.; Minaev, N.; Svidchenko, E.; Istomin, A.; Goncharuk, G.; Vladimirov, L.; Grachev, A.; Zelenetskii, A.; Timashev, P.; Akopova, T. Two-Photon-Induced Microstereolithography of Chitosan-g-Oligolactides as a Function of Their Stereochemical Composition. Polymers 2017, 9 (12), 302. https://doi.org/10.3390/polym9070302). This reference was introduced in the text (see line 141).
- Please do proof reading/grammar check.
Response: Thank you, we will follow your advice.
Sincerely yours,
On behalf of all authors,
Tatiana Tolstova
Round 2
Reviewer 1 Report
All the comments have been addressed and manuscript can be accepted in present form